# Signal changes in T2-weighted MRI of liver metastases under bevacizumab—A practical imaging biomarker?

**Johannes Thüring** *, **Christiane Katharina Kuhl, Alexandra Barabasch, Lea Hitpass, Maike Bode, Nina Bünting, Philipp Bruners, Nils Andreas Krämer**

Department of Diagnostic and Interventional Radiology, RWTH Aachen University Hospital, Aachen, Germany

* jthuering@ukaachen.de

## Abstract

### Objective

The purpose of this study was to investigate signal changes in T2-weighted magnetic resonance imaging of liver metastases under treatment with and without bevacizumab-containing chemotherapy and to compare these signal changes to tumor contrast enhancement.

### Materials and methods

Retrospective analysis of 44 patients, aged 36–84 years, who underwent liver magnetic resonance imaging including T2-weighted and dynamic contrast enhancement sequences. Patients received bevacizumab-containing (n = 22) or conventional cytotoxic chemotherapy (n = 22). Magnetic resonance imaging was obtained at baseline and at three follow-ups (on average 3, 6 and 9 months after initial treatment). Three independent readers rated the T2 signal intensity and the relative contrast enhancement of the metastases on a 5-point scale.

### Results

T2 signal intensity of metastases treated with bevacizumab showed a significant (p<0.001) decrease in T2 signal intensity after initial treatment and exhibit compared to conventionally treated metastases significantly (*p*<0.001 for each follow-up) hypointense (bevacizumab: 0.70 ± 0.83 before vs. -1.55 ± 0.61, -1.91 ± 0.62, and -1.97 ± 0.52; cytotoxic: 0.73 ± 0.79 before vs. -0.69 ± 0.81, -0.71 ± 0.68, and -0.75 ± 0.65 after 3, 6, and 9 months, respectively). T2 signal intensity was strongly correlated with tumor contrast enhancement (r = 0.71; *p*<0.001). Intra-observer agreement for T2-signal intensity was substantial (κ = 0.75). The agreement for tumoral contrast enhancement between the readers was considerably lower (κ = 0.39).

**Funding:** The authors received no specific funding for this work.

**Competing interests:** The authors have declared that no competing interests exist.

## Conclusion

Liver metastases exhibit considerably hypointense in T2-weighted imaging after treatment with bevacizumab, in contrast to conventionally treated liver metastases. Therefore, T2-weighted imaging seems to reflect the effect of bevacizumab.

## Introduction

Established image-based criteria for response assessment of solid tumors following systemic cytotoxic chemotherapy are mainly based on changes of tumor diameter (i.e. Response Evaluation Criteria in Solid Tumors [RECIST]) [1–4]. Modern chemotherapy regimens, however, often include antiangiogenic drugs, such as bevacizumab, a monoclonal antibody against vascular endothelial growth factor (VEGF) [5]. Bevacizumab primarily inhibits the formation of new blood vessels and leads to a regression of already existing immature tumor vascularization. A solid body of scientific literature indicates that these anti-angiogenic effects of bevacizumab do not cause a decrease in tumor size [6–8]. Consequently, commonly used size-based response assessment, especially RECIST, can fail to precisely monitor the effect of antiangiogenic therapy [9, 10]. Moreover, functional and often impractical as well as still metrics-based criteria, e.g., quantitative EASL (qEASL) [11], have several deficiencies and provide no practical assessment of the tumor response in clinical routine [12]. Therefore, a crucial clinical need on practical methods for tumor response assessment persists if therapy contains antiangiogenic drugs.

Magnetic resonance imaging (MRI) provides various methods for assessing the tissue functionality [13], including diffusion weighted imaging (DWI) [14] and dynamic contrast enhancement (DCE) [15]. In multiple studies, tumor enhancement showed to directly reflect the tumor vascularization [16, 17], whereas DWI does not exclusively reflect the vascularization but influenced by many factors [18, 19].

We observed, that liver metastases treated with bevacizumab containing chemotherapy appear considerably hypointense on T2-weighted sequences already after initial treatment. Therefore, the aim of this study was to investigate the changes in T2 signal intensity (T2-SI) after bevacizumab-containing chemotherapy (B-CT) and cytotoxical chemotherapy (C-CT). Moreover, these changes were compared to changes of tumor enhancement on DCE sequences, as an established functional imaging biomarker.

## Materials and methods

This institutional review board–approved retrospective study (Independent Ethics Committee of the RWTH Aachen University; EK 105/17) was conducted at an academic comprehensive cancer center and written informed consent was waived.

### Patients, target lesions, and tumor size

44 consecutive patients with a total of 67 liver metastases (26 patients with 43 metastases of colorectal cancer (CRC) and 18 patients with 24 metastases of breast cancer) were examined with standardized liver MRI from July 2010 to November 2016.

MRI was performed at baseline prior to therapy and at 3 follow-ups (FU) (average time 3, 6 and 9 months after initial treatment) under standard of care systemic therapy. Half of this cohort (n = 22) underwent B-CT, whereas the remaining patients (n = 22) underwent C-CT.

Per patient, up to 3 liver metastases were investigated by one radiologist (XX blinded for review). In cases of more than 3 metastases, three target lesions (largest metastases or those metastases with the best assessability regarding motion-artifacts) in different liver segments were determined.

The sizes (mm) of the target lesions at baseline and in all FU were measured along the longest diameter by one radiologist (XX blinded for review). Only patients with newly diagnosed metastases were rated as progressive, as rating of tumor response according to size-based criteria (e.g., RECIST) might fail the outcome of patients treated with bevacizumab [9, 10].

## MRI protocol

MRI was performed on a clinical 1.5 Tesla scanner (Ingenia, Philips, Best, The Netherlands) using a multi-channel surface receiver coil. As part of a standardized pulse sequence protocol, all examinations included a T2-weighted turbo spin echo sequence and a DCE series; further details are given in Table 1. For the DCE examinations 0.1 mmol/kg body weight gadobutrol (Gadovist, Bayer Schering Pharma, Leverkusen, Germany) was intravenously administered and images were acquired in pre-contrast, arterial, portal venous and late phase.

## T2-signal intensity assessment

T2-SI was assessed twice, in a practical reader-based (qualitative) and in a region-of-interest (ROI)-based (semi-quantitative) manner for reference purpose.

For reader-based assessment of T2-SI, three radiologists (all more than 3 years of experience in oncologic liver MRI: XX, XX, XX blinded for review) rated the signal intensity of the referring liver lesion on a 5-point scale from -2 (clearly hypointense compared to the splenic parenchyma), over 0 (isointense to the splenic parenchyma) to +2 (clearly hyperintense compared to the splenic parenchyma). In case of a disagreement in the rating of a lesion, a consensus reading carried out. The spleen was chosen as reference tissue, as the signal intensity of the liver parenchyma may change due to side effects of systemic chemotherapy.

A fourth radiologist (XX blinded for review) assessed the T2-SI of liver metastases within a representative area of the tumor excluding cystic transformed or necrotic tissue. The measured

**Table 1. MRI sequence parameters.**

| Typ of scanner | 1,5-T Ingenia, Philips Healthcare | |
|---|---|---|
| Surface coil | Multielement 16-channel coil (Sense Torso XL) | |
| | T2-weighted pulse sequence | Dynamic series |
| Pulse sequence typ | 2D turbo spin echo | T1-weighted 3D gradient echo |
| TR/TE [ms] | 2500/80 | 4.3/1.3 |
| Orientation | transverse | transverse |
| Acquisition matrix | 304 x 233 | 268 x 174 |
| Field of view | 310 mm | 330 mm |
| Slice thickness | 5 | 6 |
| Breath compensation | Respiratory triggering; in case of motion artefacts additionally breath-hold | Breath-hold |
| Sense factor | 1.4 | 2 |
| Dynamic phases | n.a. | pre-contrast, arterial, portal-venous, and equilibrium phase |

signal intensities of the metastases were intra-individually referenced to the signal intensity of the spleen by measuring round shaped ROIs on the same image.

### Analysis of contrast enhancement

Tumoral enhancement of target lesions was qualitatively evaluated by three radiologists using a 5-point scale ranging from 1 (no visible arterial or late contrast enhancement) to 5 (exception arterial or late contrast enhancement). In case of a disagreement in the rating of a lesion, a consensus reading carried out.

### Statistical analysis

Continuous variables are expressed as mean values ± standard deviation. The Spearman correlation coefficient ($\rho$) assessed the degree of monotonic association between T2-SI and tumor enhancement. Interobserver agreement between the three blinded radiologists regarding T2-SI and tumor enhancement was evaluated by using Fleiss' kappa ($\kappa$). $\kappa$ was categorized according to Landis and Koch [20].

Student's t-test was used to detect statistically significant differences in the outcomes of the investigated imaging parameters (paired t-test for intergroup (1. FU, 2. FU, and 3. FU vs. baseline) and unpaired t-test for intergroup (B-CT vs. C-CT) comparison. All test results were analyzed in an explorative way, thus p values of $p \leq 0.05$ were regarded as statistically significant. The interpretation of Spearman's $\rho$ followed the guidelines according to Altman [21]. Statistical calculation was carried out on standard PC with IBM SPSS statistics V22 (SPSS Inc., IBM, New York, USA).

## Results

### Patients, target lesions, and tumor size

Forty-four patients (28 females, 16 males) were evaluated. Patient ages ranged from 36 to 84 years (63 ± 11). The patient cohort included 26 cases of CRC as primary tumor and 18 cases of breast cancer. In total 67 consecutive target lesions were assessed (average number of lesions per patient: 1.52). Table 2 gives detailed information of the metastatic location.

Both groups (B-CT and C-CT) did not differ regarding the distribution of age, gender primary cancer, number of progressive diseases and average time between follow-up examinations (Table 3).

The average size of the metastases increased continuously from baseline to the 3. FU in both groups. For B-CT, the metastasis size change was significant between baseline vs. 1. FU, 2. FU, and 3. FU, respectively. No significant differences in tumor size were found in the C-CT group during the course of observation. In an intergroup comparison (Table 4), no significant differences in tumor size were observed.

**Table 2. Location of liver metastases.**

| Type of Primary | Number of left hepatic metastases (n) | | | | | Number of right hepatic metastases (n) | | | |
|---|---|---|---|---|---|---|---|---|---|
| | I | II | III | IVa | IVb | V | VI | VII | VIII |
| Colorectal Cancer | 1 | 3 | 3 | 5 | 6 | 6 | 5 | 6 | 8 |
| Breast Cancer | 1 | 1 | 3 | 4 | 5 | 2 | 1 | 4 | 3 |

Location of liver metastases according to hepatic segmentation by Couinaud classification.

**Table 3. Demographic and oncological details of the study population.** Of note, no significant differences between both groups were documented.

| | Study Population (n = 44) | | | | |
|---|---|---|---|---|---|
| | **Chemotherapy** | **/** | **Bevacizumab-containing chemotherapy** | **Cytotoxic-chemotherapy** | ***p*-values** |
| Cohort | Patients | n | 22 | 22 | / |
| | Number of lesions | n | 33 | 34 | / |
| Demographic details | Age | y | 65 ± 11 | 61 ± 11 | 0.482 |
| | Gender | m | 13 | 15 | 0.493 |
| | | f | 9 | 7 | 0.542 |
| Type of Primary | Number of patients with colorectal cancer | n | 13 | 13 | 1.0 |
| | Number of colorectal metastases | n | 22 | 21 | 0.964 |
| | Number of patients with breast cancer | n | 9 | 9 | 1.0 |
| | Number of breast cancer metastases | n | 11 | 13 | 0.753 |
| Patients with progressive disease | 1. follow up | n | 0 | 0 | / |
| | 2. follow up | n | 7 | 8 | 0.757 |
| | 3. follow up | n | 15 | 13 | 0.542 |
| Time of imaging | Average time in days between MRI | n | 102 ± 25 | 86 ± 24 | 0.132 |
| | | | 94 ± 22 | 86 ± 23 | 0.249 |
| | | | 88 ± 21 | 89 ± 20 | 0.867 |
| Additional agents in chemotherapy | Agents (colorectal cancer) | / | 5-Fluorouracil | Oxaliplatin | / |
| | | | Irinotecan | Trifluridine | |
| | | | Capecitabine | 5-Fluorouracil | |
| | | | | Irinotecan | |
| | | | | Capecitabine | |
| | Agents (breast cancer) | / | Paclitaxel | Doxorubicin | / |
| | | | | Epirubicin | |
| | | | | Paclitaxel | |
| | | | | 5-fluorouracil | |
| | | | | Cyclophosphamide | |
| | | | | Carboplatin | |

Average size of liver metastases at baseline, first follow-up (1. FU), second follow-up (2. FU), and third follow-up (3. FU) in the Bevacizumab-containing chemotherapy (B-CT) group and the cytotoxic chemotherapy (C-CT) group.

## Changes in T2-SI

After treatment with B-CT, the average reader-based T2-SI was significant lower in all three FU's compared to the baseline (0.70 ± 0.83 before vs. -1.55 ± 0.61, -1.91 ± 0.62, and -1.97 ± 0.52 after treatment, respectively). Comparing the T2-SI of metastases treated with C-CT, no statistical were found between baseline and the follow-up examinations (0.73 ± 0.79 before vs. -0.69 ± 0.81, -0.71 ± 0.68, and -0.75 ± 0.65 after treatment, respectively) (Table 4).

Comparing the two groups (B-CT vs. C-CT), the T2-SI did not differ at baseline ($p > 0.136$). At all three FU, the T2-SI of the metastases from the B-CT was significantly lower than in the C-CT group (Fig 1). Regarding the inter-reader reliability of agreement for T2-SI, $\kappa$ was substantial with 0.75.

ROI-based assessment of T2-SI revealed corresponding and significant ($p < 0.001$) results for every FU compared to the baseline: B-CT (0.83 ± 0.20 before vs. 0.58 ± 0.16, 0.48 ± 0.14, and 0.40 ± 0.11 after treatment, respectively), and C-CT (0.82 ± 0.22 before vs. 0.91 ± 0.21, 0.89 ± 0.19, and 0.91 ± 0.16 after treatment, respectively) (Fig 2). In an intergroup comparison (B-CT vs. C-CT), the T2-SI did not differ at baseline ($p > 0.769$). At all three FU, the T2-SI of the metastases from the B-CT group was significantly lower than in the C-CT group (Table 4).

**Table 4. Changes of liver metastases in size, T2 signal intensity, contrast enhancement.**

| | Baseline | 1. follow up | 2. follow up | 3. follow up | *p*-values (1. follow up vs. baseline) | *p*-values (2. follow up vs. baseline) | *p*-values (3. follow up vs. baseline) |
|---|---|---|---|---|---|---|---|
| **Average size (mm) of liver metastases** | | | | | | | |
| B-CT group | 15.27 | 19.80 | 22.37 | 26.43 | 0.001 | 0.001 | 0.001 |
| | ± 11.41 | ± 13.48 | ± 14.62 | ± 14.78 | | | |
| C-CT group | 15.88 | 18.34 | 19.62 | 20.63 | 0.331 | 0.101 | 0.056 |
| | ± 9.11 | ± 9.46 | ± 12.86 | ± 12.70 | | | |
| *p*-values (B-CT vs. C-CT) | 0.811 | 0.613 | 0.423 | 0.163 | | | |
| **Average reader-based metastatic T2 signal intensity** | | | | | | | |
| B-CT group | -0.70 | -1.55 | -1.91 | -1.97 | 0.001 | 0.001 | 0.001 |
| | ± 0.83 | ± 0.61 | ± 0.62 | ± 0.52 | | | |
| C-CT group | -0.73 | -0.39 | -0.33 | -0.39 | 0.072 | 0.058 | 0.063 |
| | ± 0.79 | ± 0.81 | ± 0.68 | ± 0.65 | | | |
| *p*-values (B-CT vs. C-CT) | 0.136 | 0.001 | 0.001 | 0.001 | | | |
| **Average ROI-based metastatic T2 signal intensity** | | | | | | | |
| B-CT group | 0.83 | 0.58 | 0.48 | 0.4 | 0.001 | 0.001 | 0.001 |
| | ± 0.2 | ± 0.16 | ± 0.14 | ± 0.11 | | | |
| C-CT group | 0.82 | 0.91 | 0.89 | 0.91 | 0.052 | 0.065 | 0.061 |
| | 0.22 | ± 0.21 | ± 0.19 | ± 0.16 | | | |
| *p*-values (B-CT vs. C-CT) | 0.769 | 0.001 | 0.001 | 0.001 | | | |
| **Average tumor contrast enhancement** | | | | | | | |
| B-CT group | 2.42 | 1.58 | 1.48 | 1.42 | 0.001 | 0.001 | 0.001 |
| | ± 0.78 | ± 0.74 | ± 0.66 | ± 0.65 | | | |
| C-CT group | 2.48 | 2.36 | 2.42 | 2.33 | 0.432 | 0.702 | 0.454 |
| | ± 0.69 | ± 0.75 | ± 0.68 | ± 0.65 | | | |
| *p*-values (B-CT vs. C-CT) | 0.696 | 0.001 | 0.001 | 0.001 | | | |

bevacizumab-containing chemotherapy (B-CT); cytotoxical chemotherapy (C-CT).

## Changes in tumor contrast enhancement

On average, metastases treated with B-CT exhibited less tumor enhancement at all FUs; at the 2. FU and 3. FU the differences were significant compared to baseline (2.42 ± 0.78 before vs. 1.58 ± 0.74, 1.48 ± 0.66, and 1.42 ± 0.65 after treatment, respectively). In the C-CT group, no relevant change of contrast enhancement was observed in the DCE before versus after treatment (2.48 ± 0.69 before vs. 2.36 ± 0.75, 2.42 ± 0.68, and 2.33 ± 0.65 after treatment, respectively).

The intra-group comparison revealed no statistical differences before treatment ($p > 0.696$). At all follow-ups, the B-CT showed a significantly ($p < 0.001$) decreased tumor enhancement (Table 4). With $\kappa = 0.39$, the inter-reader agreement of tumor enhancement on DCE sequences was rated as fair.

## Correlation of imaging biomarkers

The reader-based T2-SI assessment by three radiologists and the ROI-based measurements showed an excellent correlation of r = 0.91.

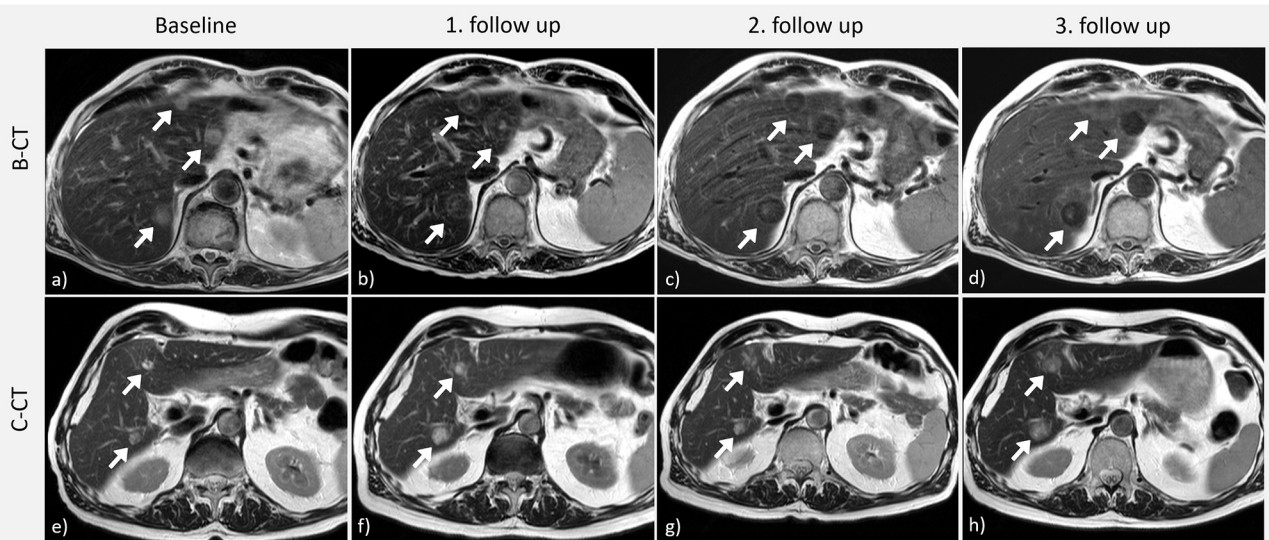

**Fig 1. Signal changes of liver metastases after bevacizumab-containing chemotherapy and cytotoxic chemotherapy.** a-d) T2-weighted MRI of a 56-year old man with hepatic metastases of rectal cancer. The treatment contained bevacizumab (B-CT). Patient was progressive with new liver metastases at the 3. follow up. 3 metastases are marked each with a white arrow. Please note the remarkable decrease in T2-signal intensity after bevacizumab therapy although slightly progress of steatosis hepatis. e-h) T2-weighted MRI of a 68-year old man with hepatic metastases of rectal cancer. Treatment did not contain bevacizumab (C-CT). Patient was progressive with new liver metastases at the 3. follow up.

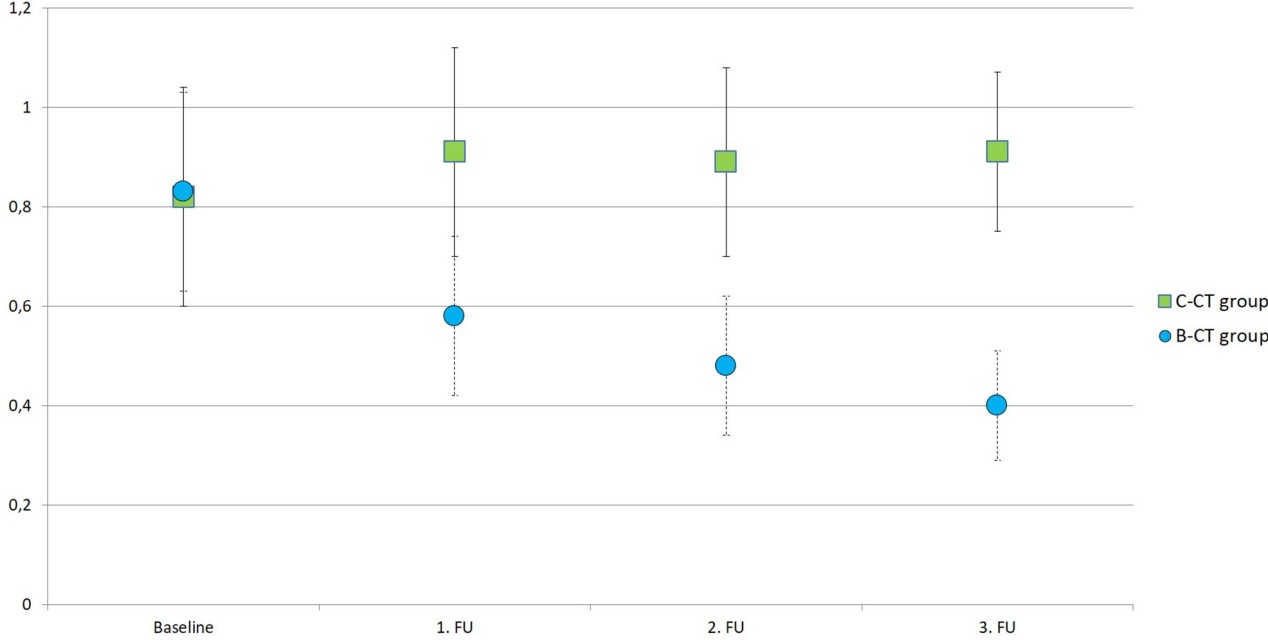

**Fig 2. Longitudinal signal changes on T2-weighted imaging after bevacizumab-containing chemotherapy and cytotoxic chemotherapy.** ROI-based T2-signal intensity of liver metastases, referenced to the signal intensity of the spleen, with (B-CT; blue balls) and without (C-CT; green squares) bevacizumab containing chemotherapy. Please note the significant (both, intra- and inter-group comparison $p<0.001$, respectively) decrease in T2-signal intensity of liver metastases after initial bevacizumab containing chemotherapy.

Post-therapeutic changes of reader-based and ROI-based assessment of T2-SI and tumor enhancement showed a strong correlation with $r = 0.71$ and $r = 0.65$, respectively.

Tumor size did not significantly correlate with T2-SI, neither ROI- nor reader-based assessment ($r = -0.05$, $p = 0.448$; $r = -0.10$, $p = 0.111$), or tumor enhancement ($r = -0.03$, $p = 0.628$).

## Discussion

Our most important finding is that liver metastases treated with bevacizumab containing systemic therapies present hypointense on T2-weighted MRI, whereas conventionally, cytotoxically treated metastases did not show relevant changes in their T2-SI.

In solid tumors, neoangiogenesis leads to an increased intra-tumoral microcirculation, tortuous micro-vessels and irregular blood flow with unstable rheology [22, 23]. Pathological examinations showed that angiogenesis inhibitors normalize the tumor vasculature, reducing the oncotic pressure, and hence normalize the interstitial water content [24]. In line to this study, it is likely that the longitudinal decrease of the T2-SI in bevacizumab treated metastases (B-CT) found in our study, may be caused by the antiangiogenic effect. In metastases that were treated cytotoxically, we did not observe such a decrease in T2-SI. This can be seen in accordance with a pathologic study where Ribero et al. [25] found significantly more vascular and sinusoidal dilation in cytotoxically treated tumors. In line with this, Shindoh et al. [4] reported a different morphological response in computed tomography of liver metastases after treatment with versus without bevacizumab containing therapies. They found that B-CT treated metastases more frequently turned into homogenous, well margined tumors (47% vs. 12%). This effect may correspond to the changes that we observed on T2-weighted MRI. However, MRI is suitable to detect distinct biopathological of the tissue and by that deriving a broader spectrum of information. Both effect–decreasing T2-SI and morphological pronounced delineation against the liver tissue on CT–might reflect the same pathological changes in the long term. Regarding the quantification of these effects, however, MRI is considered to be superior to other imaging techniques in the detection of biological changes. If a decreasing T2-SI might depict the biological changes more accurate should be further investigated.

In this intraindividual study, we investigated the behavior of two tumor entities with different vascularization characters, that are portal-venously supplied colon and arterially supplied breast metastases. Surprisingly, both entities are treated with bevacizumab containing chemotherapies following the guidelines. As the presence or absence of bevacizumab is the major aspect that dichotomizes the study cohort, and as these two groups significantly differ in their presentation on T2w MRI after therapy, we deduce that the described observations can be attributed to the effects of bevacizumab. However, it remains unclear whether the degree of T2-SI decrease is able to predict a positive outcome on the survival of the patient. Consequently, further studies should be conducted to identify the predictive value of this possible biomarker. Generally, changes in tumor vascularity are radiologically assessed using dynamic contrast enhanced imaging [4]. Thus, functional MRI techniques were examined to assess the effects of bevacizumab [26, 27]. In liver metastases, Detsky et al. [28] found a significant decrease in permeability and blood volume on perfusion MRI. In this study, we also found significantly reduced enhancement on DCE in lesions treated with bevacizumab compared to exclusive cytotoxic systemic therapies. However, recent literature claims a lack of standardization regarding the assessment of functional parameters on complex and ambiguous images as well as the extensive mandatory post-processing [29, 30]. In our study, the reader-based contrast enhancement agreed only fairly between the three observers ($\kappa = 0.39$). Although tumor enhancement is a morphologically used response criterion, the

degree of enhancement is difficult to measure and therefore less reliable [19]. Moreover, colorectal liver metastases are considered hypo-vascular tumors [31, 32] and changes of tumor perfusion are even more difficult to assess. In contrast to this, the practical reader-based T2-SI assessment on T2 turbo-spin echo sequences revealed a substantial interobserver agreement ($\kappa$ = 0.75). Underlining the validity of this reader-based assessment, the reader- and ROI-based measurement of T2-SI almost perfectly correlated (r = 0.91). In contrast to this, the quality of DCE measurements highly depends on individual patient related factors such as circulation, patient breathing, or motion. Thus, it seems to be easier and more practical to assess the signal change on T2-weighted TSE sequences rather than comparing the changes in DCE. Despite, T2-weighted imaging is not directly related to antiangiogenic effects, the changes in T2-SI most likely correlate with a therapeutic effect of bevacizumab. Additionally, this might even be of advantage, as not only the antiangiogenic effect, as in DCE, but its overall changes in the tumor metabolism and extracellular matrix are monitored.

As another functional MRI parameter, DWI has been extensively proven to serve as a technique to assess tumor response [18]. In this study, we did not compare the T2-SI changes with the DWI or the apparent diffusion coefficient (ADC), respectively. Although, similar to T2-SI, the DWI contrast/ADC are not directly related to the tumor micro-vascularity and the antiangiogenic drug effects are only indirectly reflected. However, other than T2-weighted TSE-imaging, DWI is a prone MR technique and therefore its interpretation often remains ambiguous.

The limitations of this study are its retrospective nature and the use of a DCE (4 contrast phases) instead of an over-time highly resolved perfusion technique. Moreover, respiratory artefacts may have interfered with the image evaluation, especially on the DCE images; the T2-weighted images were at least performed twice in case insufficient image quality. However, no patient had to be excluded because of minor image quality.

Although, we could show that there are distinct changes in liver metastases after treatment with bevacizumab, these results are limited in the implications for the clinical outcome. Due to the though longitudinal but retrospective setting of this study, we are not able to investigate these changes against the background of a possible change in survival. However, the benefits of a bevacizumab containing therapy are well known [6–8], and out of this our results should elucidate the effects of bevacizumab with possible practical implications on the reporting for radiologist as these changes might have an influence on the diagnostic workflow after bevacizumab treatment.

Another limitation addresses the imbalance of population size against the background of a long enrollment period (~7 years). Over this period, possible changes in patient treatment might have influenced the treatment of liver metastases. The study was designed as a longitudinal intraindividual observation within a period of up to 9 moth per patient. Moreover, no patient with any interventional tumor treatment was analyzed and a possible infestation of the metastatic rheology (e.g. portal venous embolization) was prevented within this highly controlled setting. But differences between the individual chemotherapeutical regimens during may be considered as a possible limitation.

This study shows that liver metastases treated with bevacizumab containing therapies appear hypointense on T2-weighted images. These T2 signal changes highly correlate with a reduced tumor enhancement that has been shown to reflect bevacizumab therapy effects. Towards practical efforts, reader- and ROI-based measurement of T2-SI showed a perfect correlation. Thus, T2-SI changes may serve as a functional imaging biomarker if liver metastases were treated with bevacizumab. However, future studies should evaluate the predictive value of these T2-SI changes on the clinical outcome.

## Supporting information

**S1 Data. Raw data of the patients with age, tumor type, and ROI-based measurements.**
(XLSX)

## Acknowledgments

The project was supported by the „START-Program" of the Faculty of Medicine of the RWTH Aachen University. The authors gratefully thank Daniel Truhn and Alexander Ciritsis for careful assistance during this investigation.

## Author Contributions

**Conceptualization:** Johannes Thüring, Christiane Katharina Kuhl, Philipp Bruners.

**Data curation:** Lea Hitpass, Nils Andreas Krämer.

**Formal analysis:** Johannes Thüring, Alexandra Barabasch, Lea Hitpass, Nils Andreas Krämer.

**Investigation:** Johannes Thüring, Maike Bode, Nina Bünting.

**Methodology:** Johannes Thüring, Alexandra Barabasch.

**Project administration:** Nils Andreas Krämer.

**Resources:** Lea Hitpass.

**Software:** Lea Hitpass, Nina Bünting.

**Supervision:** Christiane Katharina Kuhl, Philipp Bruners, Nils Andreas Krämer.

**Validation:** Alexandra Barabasch, Philipp Bruners.

**Visualization:** Johannes Thüring, Maike Bode, Nina Bünting.

**Writing – original draft:** Johannes Thüring.

**Writing – review & editing:** Johannes Thüring, Philipp Bruners, Nils Andreas Krämer.

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
