## [Decision Letter · Decision Letter 0]

2 Jan 2020

PONE-D-19-27984

Signal changes in T2-weighted MRI of liver metastases under Bevacizumab‒ A practical imaging biomarker ? ‒

PLOS ONE

Dear Dr. med. Thüring,

Thank you for submitting your manuscript to PLOS ONE. After careful consideration, we feel that it has merit but does not fully meet PLOS ONE’s publication criteria as it currently stands. Therefore, we invite you to submit a revised version of the manuscript that addresses the points raised during the review process.

Please address all objections and suggestions for improvement brought forward by the reviewers. If some of these issues cannot be changed improved, then please discuss the reasons comprehensively.

We would appreciate receiving your revised manuscript by Feb 16 2020 11:59PM. To enhance the reproducibility of your results, we recommend that if applicable you deposit your laboratory protocols in protocols.io, where a protocol can be assigned its own identifier (DOI) such that it can be cited independently in the future. For instructions see: http://journals.plos.org/plosone/s/submission-guidelines#loc-laboratory-protocols

We look forward to receiving your revised manuscript.

Kind regards,

Michael C Burger, M.D.

Academic Editor

PLOS ONE

Journal Requirements:

2. We note you have included a table to which you do not refer in the text of your manuscript. Please ensure that you refer to Table 2 and in your text; if accepted, production will need this reference to link the reader to the Table.

Reviewers' comments:

Reviewer's Responses to Questions

**Comments to the Author**

1. Is the manuscript technically sound, and do the data support the conclusions?

Reviewer #1: No

Reviewer #2: Yes

Reviewer #3: Partly

2. Has the statistical analysis been performed appropriately and rigorously? 

Reviewer #1: Yes

Reviewer #2: Yes

Reviewer #3: Yes

3. Have the authors made all data underlying the findings in their manuscript fully available?

Reviewer #1: Yes

Reviewer #2: Yes

Reviewer #3: No

4. Is the manuscript presented in an intelligible fashion and written in standard English?

Reviewer #1: Yes

Reviewer #2: Yes

Reviewer #3: Yes

5. Review Comments to the Author

Reviewer #1: In this study, the authors analyzed the impact of T2-weighted MRI on liver metastases from colorectal and breast primary treated with or without anti-VEGF antibody therapy. The conclusion is that T2 weighted imaging can reflect the effect of bavacizumab.

I have some comments.

1. As mentined in the discussion, the tumor regression after anti-VEGF therapy is unique that is called "morphological change" on CT reported from MDACC group. Basically, you only validated the morphological change on MRI, didn't you? You should more emphasize the superiority of this study over the previous CT-based studies because CT is more universally diffused.

2.You did not show the clincal outcome of the patients. I did not know the correlation of the change on the MRI with the clincal outcome. You need to analyze that if you want to show the benefit of T2-weighted MRI as a new biomarker.

Reviewer #2: The manuscript is well written, easy to understand and the results are clearly explained. The discussion clearly correlate the results with current litterature. For this reason I suggest that it can be published in the current form

Reviewer #3: The finality of this study was to investigate signal changes in T2-weighted magnetic resonance

imaging of liver metastases under treatment with and without bevacizumab-containing

chemotherapy and to compare these signal changes to tumor contrast enhancement.

It is true that this is a crucial clinical need on usual methods for tumor response

assessment persists if therapy contains antiangiogenic drugs like bevacizumab, TKI inhibitors, and others drugs in many type of tumors where now these modern drugs are used.

It is well known that many modern drugs and Bevacizumab, the first one adopted worldwidely in chemotherapy, primarily inhibits the formation of new blood vessels and leads to a regression of already existing tumor vascularization. Obviously a great debate is still open about

commonly used size-based response assessment, especially RECIST, can fail to precisely

monitor the effect of antiangiogenic therapy.

The basic idea of the Authors is that liver metastases exhibit considerably hypointense in T2-weighted imaging after treatment with bevacizumab, in contrast to conventionally treated liver metastases. Therefore, T2-weighted imaging seems to reflect the effect of bevacizumab

After sharing these good intentions of the Authors, it is necessary to point out some conceptual problems that seriously undermine many of the reported observations and the good work done.

First of all the Authors present a limited number of Patients that received bevacizumab-containing (n=22) or conventional cytotoxic chemotherapy (n=22). Again the recruitment is too long and in these 7 years many aspects are changed. A serious conceptual error is to consider colon cancer, known to be hypovascular, and breast cancer which are very well vascularized.

Another methodological error is to consider few metastases per patient, it is known that after chemotherapy with or without angiogenic agents the response varies in the various metastases and segments. Because 44 consecutive patients with a total of 67 liver metastases (26 patients with 43 metastases of colorectal cancer (CRC) and 18 patients with 24 metastases of breast cancer) were studied with standardized liver MRI from July 2010 to November 2016.

I agree with the finding is that liver metastases treated with bevacizumab containing

systemic therapies present hypointense on T2-weighted MRI, whereas conventionally,

cytotoxically treated metastases did not show relevant changes in their T2-SI.

As the Authors report colorectal liver metastases are considered hypo-vascular tumors and changes of tumor perfusion are even more difficult to assess.

This study present other bias: its retrospective nature and the use of a DCE (4 contrast

phases) instead of an over-time highly resolved perfusion technique.

Even if the Authors report that respiratory artefacts may have interfered with the image evaluation, especially on the DCE images, the claim that T2-weighted images were at least performed twice in case insufficient image quality is not sufficent to accept considering

minor image quality.

The conclusion that.. “T2-SI changes may serve as a functional imaging biomarker if liver

metastases were treated with bevacizumab” is not fully acceptable considering this paper.

In any case, it must be recognized that the authors' intentions are very useful in clinical practice and this work is part of a very important discussion in the evaluation of responses to chemotherapy.

The work is well written, well developed and must be reconsidered after choosing only one pathology (colon or breast) which has a homogeneous pattern of vascularization.

The lesions must be mapped for segments and at least 3 metastases per patient must be considered to have usable data and conclude as the authors say.

In conclusion, the work is well written and significant for the research it involves on a very important topic but greater conceptual changes are needed. It may be published after a major revision.

6. PLOS authors have the option to publish the peer review history of their article (what does this mean?). If published, this will include your full peer review and any attached files.

Reviewer #1: No

Reviewer #2: No

Reviewer #3: Yes: Giammaria Fiorentini

---

## [Author Response · Author response to Decision Letter 0]

14 Feb 2020

Point-by-Point Reply to the Editor’s and Reviewers’ Comments

Title: 

Signal changes in T2-weighted MRI of liver metastases under Bevacizumab

‒ A practical imaging biomarker ? ‒

Reference number: 

PONE-D-19-27984

Journal: 

PLOS ONE

General Reply:

Thank you very much for the thorough review of our manuscript. We very much appreciate the opportunity and hope to have satisfactorily answered all of your and the reviewers’ comments. We would be happy if our manuscript was chosen to be published in PLOS ONE.

In the following we would like to address the comments of the reviewers and the editor point by point. Please note, that changes to the original document are tracked with “track changes” in word. As required, we have also attached a “clean” version in which all of the changes have been accepted.

 

Reply to Reviewer #1

PONE-D-19-27984

Title: 

Signal changes in T2-weighted MRI of liver metastases under Bevacizumab

‒ A practical imaging biomarker ? ‒

Comment 1.1: “In this study, the authors analyzed the impact of T2-weighted MRI on liver metastases from colorectal and breast primary treated with or without anti-VEGF antibody therapy. The conclusion is that T2 weighted imaging can reflect the effect of bavacizumab. I have some comments.” 

Authors’ Response: We would like to thank the Reviewer very much for taking the time to review our manuscript and for her/his overall appreciation of our work.

Comment 1.2: ” 1. As mentioned in the discussion, the tumor regression after anti-VEGF therapy is unique that is called "morphological change" on CT reported from MDACC group. Basically, you only validated the morphological change on MRI, didn't you? You should more emphasize the superiority of this study over the previous CT-based studies because CT is more universally diffused.”

Authors’ Response: Indeed, these imprecisions in our discussion warrant further explanation: In our study liver metastases showed characteristically changes in their signal behaviour on T2-weighted MRI. These changes might be in line with the changes described by the MDACC group. However, MRI is suitable to detect a broader spectrum of information compared to CT. The changes described by MDACC group are related to the morphological appearance of the metastases (e.g. round-shaped). In contrast to this, we could show that the texture of the metastases exhibits changes that might reflect biopathological changes in the metastases itself. Moreover, this has important implications for the radiologist in the reporting workflow of patients treated with antiangiogenic therapy. In general, metastases exhibit an intermediate-to-hyperintense signal on T2-weighted imaging. Nevertheless, with this publication we want to make radiologist aware of these changes after treatment with bevacizumab to maintain the diagnostic accuracy in these cases. 

So, in line with the reviewer’s suggestion, we added the following to the discussion on page 16 (ll. 252-258) in order to guide the astute reader in understanding our line of thought:

“However, MRI is suitable to detect distinct biopathological of the tissue and by that deriving a broader spectrum of information. Both effect – decreasing T2-SI and morphological pronounced delineation against the liver tissue on CT – might reflect the same pathological changes in the long term. Regarding the quantification of these effects, however, MRI is considered to be superior to other imaging techniques in the detection of biological changes. If a decreasing T2-SI might depict the biological changes more accurate should be further investigated.”

Comment 1.3: “2. You did not show the clincal outcome of the patients. I did not know the correlation of the change on the MRI with the clincal outcome. You need to analyze that if you want to show the benefit of T2-weighted MRI as a new biomarker.”

Authors’ Response: We totally agree with the reviewer’s criticism. However, this study was initiated to describe MR imaging effects that seem to be typical for bevacizumab containing chemotherapies and might further serve as an additional parameter in follow-up exams. The effects of bevacizumab regarding the patient outcome after tumor treatment are well studied and were not part of this investigation. This study was accomplished with a longitudinal and retrospective approach; therefore, we are not able to investigate whether our findings have an implication on the clinical outcome, although we would like to. As this a critical limitation, we added this to the limitations on page 17 (ll. 305-312):

“Although, we could show that there are distinct changes in liver metastases after treatment with bevacizumab, these results are limited in the implications for the clinical outcome. Due to the though longitudinal but retrospective setting of this study, we are not able to investigate these changes against the background of a possible change in survival. However, the benefits of a bevacizumab containing therapy are well known (1-3), and out of this our results should elucidate the effects of bevacizumab with possible practical implications on the reporting for radiologist as these changes might have an influence on the diagnostic workflow after bevacizumab treatment.“

 

Reply to Reviewer #2

PONE-D-19-27984

Title: 

Signal changes in T2-weighted MRI of liver metastases under Bevacizumab

‒ A practical imaging biomarker ? ‒

Comment 2.1: “The manuscript is well written, easy to understand and the results are clearly explained. The discussion clearly correlate the results with current litterature. For this reason I suggest that it can be published in the current form.”

Authors’ Response: Thank you very much for your overall appreciation of our manuscript. Moreover, we agree with you and hope to find the right and interested readership in the journal PLOS ONE and hope to publish this manuscript in the journal.

 

Reply to Reviewer #3

PONE-D-19-27984

Title: 

Signal changes in T2-weighted MRI of liver metastases under Bevacizumab

‒ A practical imaging biomarker ? ‒

Comment 3.1: ” The finality of this study was to investigate signal changes in T2-weighted magnetic resonance imaging of liver metastases under treatment with and without bevacizumab-containing chemotherapy and to compare these signal changes to tumor contrast enhancement. It is true that this is a crucial clinical need on usual methods for tumor response assessment persists if therapy contains antiangiogenic drugs like bevacizumab, TKI inhibitors, and others drugs in many type of tumors where now these modern drugs are used.

It is well known that many modern drugs and Bevacizumab, the first one adopted worldwidely in chemotherapy, primarily inhibits the formation of new blood vessels and leads to a regression of already existing tumor vascularization. Obviously, a great debate is still open about commonly used size-based response assessment, especially RECIST, can fail to precisely monitor the effect of antiangiogenic therapy. The basic idea of the Authors is that liver metastases exhibit considerably hypointense in T2-weighted imaging after treatment with bevacizumab, in contrast to conventionally treated liver metastases. Therefore, T2-weighted imaging seems to reflect the effect of bevacizumab”

Authors’ Response: We would like to thank the Reviewer for his/her thorough revision of our manuscript, its overall appreciation and for the numerous comments and detailed feedback that we hope to have addressed sufficiently. We are confident that the revision of our manuscript along these lines has been of great benefit to its quality. Please refer to our detailed responses to the comments below.

Comment 3.2: ”After sharing these good intentions of the Authors, it is necessary to point out some conceptual problems that seriously undermine many of the reported observations and the good work done. First of all the Authors present a limited number of Patients that received bevacizumab-containing (n=22) or conventional cytotoxic chemotherapy (n=22). Again the recruitment is too long and in these 7 years many aspects are changed.”

Authors’ Response: We totally understand the reviewers’ concerns and with her/his criticisms of the long recruitment phase. We have to point out, that there may have been differences within the chemotherapy regimens of the patients related to ongoing improvements in oncological patient care. However, the main difference between the observed groups was the presence or absence of bevacizumab. Consequently, this study was designed as an intraindividual comparison to observe the bevacizumab effects in a longitudinal manner. As a second aspect of this reviewer’s concerns, we performed a highly standardized pulse sequence protocol, that did not change within observation period and was carried out on the same 1.5 T Philips MR scanner, so we can guarantee that MRI technique did not influence the study results. Regarding the small sample size (2 x n=22) against the background of an enrollment phase of 7 years, we have to point out, that we need to have highly comparable patients in this longitudinal intraindividual study design. No patient received any interventional tumor treatment, hemihepatectomy, portal vein embolization or other tumor related interventions. As a mater of fact, this is one of the major strengths of our study, and which is in contrast to other studies on bevacizumab (4, 5).

In line with the reviewer’s concerns, we added the following to the limitations sections (page 18; ll. 313-320) to improve the interpretation of our results by the readers: 

“Another limitation addresses the imbalance of population size against the background of a long enrollment period (~7 years). Over this period, possible changes in patient treatment might have influenced the treatment of liver metastases. The study was designed as a longitudinal intraindividual observation within a period of up to 9 moth per patient. Moreover, no patient with any interventional tumor treatment was analyzed and a possible infestation of the metastatic rheology (e.g. portal venous embolization) was prevented within this highly controlled setting. But differences between the individual chemotherapeutical regimens during may be considered as a possible limitation.” 

Comment 3.3: ”A serious conceptual error is to consider colon cancer, known to be hypovascular, and breast cancer which are very well vascularized.”

Authors’ Response: The Reviewer is right to remark this and in line with the Reviewer we have to admit that this seems intuitively sound. However, anti-angionetic drugs are applied in both highly vascularized breast cancer and hypovascular colon cancer. As this longitudinal study observes the intraindividual effects, these physiological cancer properties are eliminated. Moreover, we did not focus on dynamic contrast enhanced MRI to observe perfusion effects that are highly variable even at baseline and that can only insufficiently be monitored using often quite variable DCE sequences. Instead, we focused on the reproducible T2-weighted TSE sequence to observe possible chemotherapeutical induced effects. In line with our conception we have to admit, that first admission for bevacizumab was granted for hypovascularized colon cancer (2005) and hereafter an admission for the hypervascularized breast cancer was granted (2007). We have considered this in our initial design for this study and wanted to point out that the similar behavior in T2-weighted TSE could overcome possible masking effects in to date shortcomings of dynamic enhancement imaging. T2-weighted MRI might be suitable to depict changes of antiangiogenic therapy regardless of the initial metastatic vascularization which is significantly demonstrated in this study. To discuss this crucial issue, we added the following to the discussion section on page 17 (ll. 259-268):

“In this intraindividual study, we investigated the behavior of two tumor entities with different vascularization characters, that are portal-venously supplied colon and arterially supplied breast metastases. Surprisingly, both entities are treated with bevacizumab containing chemotherapies following the guidelines. As the presence or absence of bevacizumab is the major aspect that dichotomizes the study cohort, and as these two groups significantly differ in their presentation on T2w MRI after therapy, we deduce that the described observations can be attributed to the effects of bevacizumab. However, it remains unclear whether the degree of T2-SI decrease is able to predict a positive outcome on the survival of the patient. Consequently, further studies should be conducted to identify the predictive value of this possible biomarker.” 

Comment 3.4: ”Another methodological error is to consider few metastases per patient, it is known that after chemotherapy with or without angiogenic agents the response varies in the various metastases and segments. Because 44 consecutive patients with a total of 67 liver metastases (26 patients with 43 metastases of colorectal cancer (CRC) and 18 patients with 24 metastases of breast cancer) were studied with standardized liver MRI from July 2010 to November 2016.”

Authors’ Response: We do agree with the reviewer’s statement. Thus, we assessed all metastases, up to 3, in all cases. If the number of metastases was >3 and the lesions were spread over the liver, we selected metastases all from different segments. To address this valuable point, we clarified this in the method and results section (page 5 ll. 94-95 and 10 ll. 152-156) and added another table (table 2) which summarizes the exact location of all metastases: 

“In cases of more than 3 metastases, three target lesions (largest metastases or those metastases with the best assessability regarding motion-artifacts) in different liver segments were determined.” 

“Table 2 gives detailed information of the metastatic location.”

Table 2: Location of liver metastases 

Type of Primary Number of left hepatic metastases (n) Number of right hepatic 

metastases (n)

 I II III IVa IVb V VI VII VIII

Colorectal Cancer 1 3 3 5 6 6 5 6 8

Breast Cancer 1 1 3 4 5 2 1 4 3

Table 2: Location of liver metastases according to hepatic segmentation by Couinaud classification. 

Comment 3.5: ”I agree with the finding is that liver metastases treated with bevacizumab containing systemic therapies present hypointense on T2-weighted MRI, whereas conventionally, cytotoxically treated metastases did not show relevant changes in their T2-SI.

As the Authors report colorectal liver metastases are considered hypo-vascular tumors and changes of tumor perfusion are even more difficult to assess.”

Authors’ Response: We would like to thank the reviewer for sharing his/her considerations on the role of perfusion-related imaging in general and by this support the clinical need of a practical imaging biomarker in the daily clinical routine. 

Comment 3.6: ” This study present other bias: its retrospective nature…”

Authors’ Response: We agree, that a retrospective design is a general limitation. In line with the qualified criticism of Reviewer 1 we have addressed this issue in our limitations and by this hope to drive future prospective studies on that persisting clinical need. 

Comment 3.7: ”…the use of a DCE (4 contrast phases) instead of an over-time highly resolved perfusion technique.”

Authors’ Response: Thank you for this thoughtful remark. We agree that perfusion imaging is the state of art technique in an experiential setting. Although, perfusion is technically highly challenging and thus prone to errors. Moreover, it is challenging for the patient compliance (breath hold etc.). As a matter of fact, it is rarely part of a standard clinical examination. We strive to aim with this study for a practical imaging biomarker in everyday patient care. However, we absolutely agree with these concerns, and hope to yield the scientific basis for a possible prospective study, in which perfusion MRI should be part of the study MRI protocol. We hope that we have addressed these concerns sufficiently with our limitations section on page 17.

Comment 3.8: “Even if the Authors report that respiratory artefacts may have interfered with the image evaluation, especially on the DCE images, the claim that T2-weighted images were at least performed twice in case insufficient image quality is not sufficent to accept considering

minor image quality.”

Authors’ Response: Generally, follow-up MRI exams of the abdomen are challenging for the majority of patients. In particular, as in our population we examined patients with a crucial illness and thus optimal compliance was hard to achieve. Being part of a cancer comprehensive center with standardize examination routines, we consequently perform T2-weighted-TSE sequences with respiratory gating, and in case of minor image quality, in breath hold. Using DCE, all these challenges (shortness of breath etc.) become even more critical. However, as we performed measurements in a ROI-based manner, we were able to avoid measurements in artifacts. Against this background, we have to state - according to our discussion section (page 17) - that no patient was excluded due to insufficient image quality, neither in T2-weighted imaging nor on DCE imaging. 

Comment 3.9: “The conclusion that.. “T2-SI changes may serve as a functional imaging biomarker if liver metastases were treated with bevacizumab” is not fully acceptable considering this paper. In any case, it must be recognized that the authors' intentions are very useful in clinical practice and this work is part of a very important discussion in the evaluation of responses to chemotherapy.”

Authors’ Response: We share the intention of the reviewer and in line with his astute criticism and the considerations of Reviewer #1 we hope to sufficiently address this issue in the discussion section. On and above, we would like to point out that our results underpin the highly published results of Shindoh and Chun et al. (4, 5). The results we would like to publicize with this manuscript are the first descriptions of changes in MRI after bevacizumab therapy. These changes in T2-weighted MRI are correlated to some extend to the perfusion and consecutively to the rheology of the metastases, which is in line with the above-mentioned publications. By increasing use of antiangiogenic drugs, radiologist should be aware of these changes in everyday patient care. We do not strive to overvalue the implications of T2-weighted imaging in a context of a deeper pathological tumorbiological response. We hope to donate certain guidance to the readers through this meaningful scientific discussion on antiangiogenetic therapy by our discussion. 

Comment 3.10: “The work is well written, well developed and must be reconsidered after choosing only one pathology (colon or breast) which has a homogeneous pattern of vascularization.”

Authors’ Response: We would like to thank the reviewer for her/his commendation. Regarding focusing on only one entity, we would like to refer to our reply of comment 3.3 and its consequences. We still agree, that only one entity seems more intuitive in the context of an interindividual setting, however we could show that the observed effect can be attributed to bevacizumab and are independent form the tumor entity and its initial stet of perfusion in our intraindividual longitudinal observation.

Comment 3.11: “The lesions must be mapped for segments and at least 3 metastases per patient must be considered to have usable data and conclude as the authors say.”

Authors’ Response: In line with the reviewer’s suggestion we provided a detailed mapping (table 2; pls. be referred to authors’ response to comment 3.4) of the metastases with the location of all metastases and an exact distribution of all in the segments. 

Comment 3.12: “In conclusion, the work is well written and significant for the research it involves on a very important topic but greater conceptual changes are needed. It may be published after a major revision.”

Authors’ Response: Thank you for this kind comment. We hope to have replied to your entire satisfaction, and we further hope, that you esteem this publication worthy to be published.  

Reply to the Academic Editor

PONE-D-19-27984

Title: 

Signal changes in T2-weighted MRI of liver metastases under Bevacizumab

‒ A practical imaging biomarker ? ‒

Comment AE 1: “Thank you for submitting your manuscript to PLOS ONE. After careful consideration, we feel that it has merit but does not fully meet PLOS ONE’s publication criteria as it currently stands. Therefore, we invite you to submit a revised version of the manuscript that addresses the points raised during the review process." 

Authors’ Response: Thank you very much for your overall appreciation of our manuscript. We hope to have addressed all issues raised by you and the reviewers. Moreover, we agree with you and hope to find the right and interested readership in the journal PLOS ONE and keen on publishing this manuscript in the journal.

Comment AE 2: “When submitting your revision, we need you to address these additional requirements: 

1. Please ensure that your manuscript meets PLOS ONE's style requirements, including those for file naming. The PLOS ONE style templates can be found at http://www.plosone.org/attachments/PLOSOne_formatting_sample_main_body.pdf and http://www.plosone.org/attachments/PLOSOne_formatting_sample_title_authors_affiliations.pdf”

Authors’ Response: We apologize for not fulfilling the requirements of PLOS ONE style. We thoroughly revised the entire manuscript and hope to sufficiently address all formal requirements. 

Comment AE 3: “2. We note you have included a table to which you do not refer in the text of your manuscript. Please ensure that you refer to Table 2 and in your text; if accepted, production will need this reference to link the reader to the Table.”

Authors’ Response: We apologize for this lack of clear presentation. The editor is right in that the table 2 was missed to mentioned in the text. Due to the response to the reviewers another table was inserted and table 3 is now mentioned in the text, as requested. 

Comment AE 4: “3. Please include captions for your Supporting Information files at the end of your manuscript, and update any in-text citations to match accordingly. Please see our Supporting Information guidelines for more information: http://journals.plos.org/plosone/s/supporting-information.”

Authors’ Response: Captions for the Supporting Information files have been added. 

 

References:

1. Brufau BP, Cerqueda CS, Villalba LB, Izquierdo RS, González BM, Molina CN. Metastatic renal cell carcinoma: radiologic findings and assessment of response to targeted antiangiogenic therapy by using multidetector CT. Radiographics. 2013;33(6):1691-716.

2. Krajewski KM, Nishino M, Franchetti Y, Ramaiya NH, Abbeele AD, Choueiri TK. Intraobserver and interobserver variability in computed tomography size and attenuation measurements in patients with renal cell carcinoma receiving antiangiogenic therapy: implications for alternative response criteria. Cancer. 2014;120(5):711-21.

3. Smith AD, Zhang X, Bryan J, Souza F, Roda M, Sirous R, et al. Vascular Tumor Burden as a New Quantitative CT Biomarker for Predicting Metastatic RCC Response to Antiangiogenic Therapy. Radiology. 2016;281(2):484-98.

4. Shindoh J, Loyer EM, Kopetz S, Boonsirikamchai P, Maru DM, Chun YS, et al. Optimal morphologic response to preoperative chemotherapy: an alternate outcome end point before resection of hepatic colorectal metastases. Journal of Clinical Oncology. 2012;30(36):4566.

5. Chun YS, Vauthey J-N, Boonsirikamchai P, Maru DM, Kopetz S, Palavecino M, et al. Association of computed tomography morphologic criteria with pathologic response and survival in patients treated with bevacizumab for colorectal liver metastases. Jama. 2009;302(21):2338-44.

---

## [Decision Letter · Decision Letter 1]

4 Mar 2020

Signal changes in T2-weighted MRI of liver metastases under Bevacizumab

‒ A practical imaging biomarker ? ‒

PONE-D-19-27984R1

Dear Dr. Thüring,

We are pleased to inform you that your manuscript has been judged scientifically suitable for publication and will be formally accepted for publication once it complies with all outstanding technical requirements.

With kind regards,

Michael C Burger, M.D.

Academic Editor

PLOS ONE

Additional Editor Comments (optional):

Reviewers' comments:

Reviewer's Responses to Questions

**Comments to the Author**

1. If the authors have adequately addressed your comments raised in a previous round of review and you feel that this manuscript is now acceptable for publication, you may indicate that here to bypass the “Comments to the Author” section, enter your conflict of interest statement in the “Confidential to Editor” section, and submit your "Accept" recommendation.

Reviewer #1: All comments have been addressed

Reviewer #3: All comments have been addressed

2. Is the manuscript technically sound, and do the data support the conclusions?

Reviewer #1: Yes

Reviewer #3: Partly

3. Has the statistical analysis been performed appropriately and rigorously? 

Reviewer #1: Yes

Reviewer #3: Yes

4. Have the authors made all data underlying the findings in their manuscript fully available?

Reviewer #1: Yes

Reviewer #3: No

5. Is the manuscript presented in an intelligible fashion and written in standard English?

Reviewer #1: Yes

Reviewer #3: Yes

6. Review Comments to the Author

Reviewer #1: The authors have successfully responded the reviewer's comments. I think this manuscript to be worth publishing in the journal of Plos One.

Reviewer #3: It is a good job, well articulated and well written. The article has been much improved and I am delighted with the authors.

Unfortunately, however, I believe that the mistake of considering two such different pathologies as colon cancer and breast cancer is unacceptable.

If it is correct to say that bevacizumab is used in both diseases, but it is associated with very different drugs in colon cancer compared to breast cancer.

This is an insuperable conceptual mistake that would generate confusion in our readers.

My very friendly and respectful advice of the excellent work done is to re-submit the same work dedicated only to one tumor and perhaps increasing the number of cases that remains poor.

I am really very sorry to reject this very appreciable work but the conceptual and methodological mistake in evaluating in the same way such different metastases under the vascular profile cannot be published

7. PLOS authors have the option to publish the peer review history of their article (what does this mean?). If published, this will include your full peer review and any attached files.

Reviewer #1: No

Reviewer #3: Yes: Giammaria Fiorentini

---

## [Editor Report · Acceptance letter]

16 Mar 2020

PONE-D-19-27984R1 

Signal changes in T2-weighted MRI of liver metastases under Bevacizumab
‒ A practical imaging biomarker ? ‒ 

Dear Dr. Thüring:

I am pleased to inform you that your manuscript has been deemed suitable for publication in PLOS ONE. Congratulations! Your manuscript is now with our production department. 

With kind regards,

on behalf of

Dr. Michael C Burger 

Academic Editor

PLOS ONE